# High-Efficiency and Broadband Near-Infrared Bi-Functional Metasurface Based on Rotary Different-Size Silicon Nanobricks

**DOI:** 10.3390/nano9121744

**Published:** 2019-12-07

**Authors:** Wei Wang, Chong Guo, Jingluo Tang, Zehan Zhao, Jicheng Wang, Jinghua Sun, Fei Shen, Kai Guo, Zhongyi Guo

**Affiliations:** 1Department of Mathematics and Physics, Shijiazhuang Tiedao University, Shijiazhuang 050043, China; wangw@stdu.edu.cn (W.W.); guocong122@163.com (C.G.); tangjl12@126.com (J.T.); xvxvxiaozhao@163.com (Z.Z.); 2School of Electrical Engineering and Intelligentization, Dongguan University of Technology, Dongguan 523808, China; sunjh@dgut.edu.cn (J.S.); shenfei@hfut.edu.cn (F.S.); 3School of Science, Jiangnan University, Wuxi 214122, China; jcwang@jiangnan.edu.cn; 4School of Computer and Information, Hefei University of Technology, Hefei 230009, China; kai.guo@hfut.edu.cn

**Keywords:** metasurface, phase modulation, vortex beam, metalens

## Abstract

Several novel spin-dependent bi-functional metasurfaces consisting of different-sized rotary silicon nanobricks have been proposed and numerically investigated based on the Pancharatnam–Berry phase and structural phase simultaneously. Here, a transmission mechanism is strictly deduced, which can avoid crosstalk from the multiplexed bi-functional metasurface. Four kinds of high-efficiency bi-functional devices have been designed successfully at infrared wavelengths, including a spin-dependent bi-functional beam deflector, a spin-dependent bi-functional metalens, a bi-functional metasurface with spin-dependent focusing and deflection function, and a spin-dependent bi-functional vortex phase plate. All of the results demonstrate the superior performances of our designed devices. Our work opens up new doors toward building novel spin-dependent bi-functional metasurfaces, and promotes the development of bi-functional devices and spin-controlled photonics.

## 1. Introduction

In recent times, metasurfaces, usually composed of a series of periodic or regionally nonperiodic arrayed unit cells with subwavelength thicknesses, have attracted much attention in the field of electromagnetic wave manipulation, and they can effectively control polarization state, amplitude, and wavefront. For the designing of efficient metasurfaces, one basic approach is to try to achieve a phase range of 2π. The phase can be manipulated through changing the size or rotating the unit cells. Therefore, we can design the desired phase distributions by arraying the unit cells as required, and manipulate the wavefront and the polarization state of the output light. In the past few years, there have been rapid and significant developments in both theory and practical research about metasurfaces. Many fascinating phenomena have been discovered, such as anomalous reflection and refraction [1,2,3], metalenses [4,5,6], polarization manipulation [7,8,9], vortex beam generation [10,11,12], and absorbers [13]. Additionally, many ingenious bi-functional metasurfaces have been designed, which holds great promise for integrated optics.

In the field of integrated optics, functionality integration, which is the base of realizing device integration and miniaturization, has recently attracted great research interest. It is very difficult to apply conventional multifunctional devices to integrated optics due to their large size and complicated structures. In addition, metasurfaces encourage functionality integration. Bi-functional metasurfaces have been widely studied, mainly based on either structural phase metasurfaces [14,15,16,17,18,19,20,21] or Pancharatnam–Berry (PB) phase metasurfaces [22,23,24,25,26,27,28,29,30,31]. For structural phase metasurfaces, the local phases of unit cells are controlled by structural resonances through changing the structure size, but they not only possess complex structures but also only work under linearly polarized (LP) incident light. For PB phase metasurfaces, the local extra phases of unit cells are controlled by orientation angles through rotating the structure, and they exhibit great ability in manipulating circularly polarized (CP) light. However, most PB phase metasurfaces are designed by multiplexing different metasurfaces with different functions, which have intrinsic limitations and therefore lower efficiency. Recently, bi-functional metasurfaces were proposed and designed which possess different functions for transmission and reflection [32]. These bi-functional metasurfaces are all based on the structural phase or PB phase. To achieve a high-efficiency bi-functional metasurface for left circularly polarized (LCP) and right circularly polarized (RCP) light, Capasso realized different chiral holograms [33] and a vortex beam with different topological charges [34] for LCP and RCP light, based on the combination of structural phase and PB phase in the visible band. Zhao realized multichannel vectorial holograms [35] at 600–800 nm based on the same method. Meanwhile, Groever proposed a metalens [36] at 500–620 nm based on the same method, in which the numerical aperture was about 0.05, and the fractional bandwidth was about 21% (bandwidth of 120 nm and central wavelength of 560 nm). It should be noted that these papers only achieved the metasurface with a particular function based on the designing method. It is still a great challenge to design bi-functional metasurfaces with different functions over a broad band in the near-infrared region based on the combination of two kinds of phase. 

In this paper, we propose several broadband spin-dependent bi-functional metasurfaces which are constructed by rotary silicon nanobricks (RSNs) based on the combining manipulations of structural phase and PB phase simultaneously. Four kinds of bi-functional metasurfaces have been designed, which possess completely different functions and design philosophy. A bi-functional beam deflector (deflecting the LCP and RCP lights to different directions), a bi-functional metalens (focusing the LCP and RCP lights to different positions), a bi-functional metasurface for focusing and deflection, and a spin-dependent bi-functional vortex phase plate (forming different vortex beams for the LCP and RCP incidences) have been successfully designed, which demonstrate superior performances in their functions accordingly. The broadband characteristic was also explored, and the work bandwidth is about 500 nm (in the range of 1200–1700 nm). Above all, the fractional bandwidth reaches up to 34%. One bi-functional vortex phase plate was also designed, allowing the LCP and RCP incident light to produce a vortex beam with different topological charges. These spin-dependent bi-functional metasurfaces are significant for the development of integrated optics, which can realize the bi-functional devices without crosstalk.

## 2. Theoretical Analysis

To present a bi-functional metasurface without functional crosstalk, we firstly try to analyze and study the dielectric anisotropic rotary nanobrick, which can introduce elastic phase delays for LCP and RCP incidences. Generally speaking, for a nanobrick, the transmission matrix *T* [11] can be written as
(1)$$T=\left(\begin{array}{l} \begin{array}{cc} T_{xx} & T_{xy} \end{array}\\ \begin{array}{cc} T_{yx} & T_{yy} \end{array} \end{array}\right)$$
where $T_{xy}$ is the complex transmission coefficient for the x-linearly polarized (XLP) transmission from the incident y-linearly polarized (YLP) light, and all of the other elements possess similar definitions. Because there is basically no coupling between the x-polarization and y-polarization, it leads to $T_{xy}=T_{yx}=0$. Because the rotary nanobrick possesses a structural phase depending on the size of the nanobrick, the complex transmission coefficient can be specifically expanded in the form of amplitudes and phases, and the transmission matrix T can be rewritten as
(2)$$T=\left(\begin{array}{cc} \begin{array}{c} t_{xx}e^{i\varphi_{xx}}\\ 0 \end{array} & \begin{array}{c} 0\\ t_{yy}e^{i\varphi_{yy}} \end{array} \end{array}\right)$$
where $t_{xx}$ and $\varphi_{xx}$ express the XLP transmission coefficient and structural phase, and $t_{yy}$ and $\varphi_{yy}$ express the YLP transmission coefficient and structural phase, respectively, which depend heavily on the geometry and size of the nanobrick. For a nanobrick rotated by an angle of θ, as shown in Figure 1a, the new Jones matrix can be expressed as
(3)$$\begin{array}{l} T^{*}=R(-\theta )\left(\begin{array}{l} \begin{array}{cc} t_{xx}e^{i\varphi_{xx}} & 0 \end{array}\\ \begin{array}{cc} 0 & t_{yy}e^{i\varphi_{yy}} \end{array} \end{array}\right)R(\theta )\\ =\frac{1}{2}\left(\begin{array}{ll} t_{xx}e^{i\varphi_{xx}}+t_{yy}e^{i\varphi_{yy}}+\left(t_{xx}e^{i\varphi_{xx}}-t_{yy}e^{i\varphi_{yy}}\right)\cos2\theta & (t_{xx}e^{i\varphi_{xx}}-t_{yy}e^{i\varphi_{yy}})\sin2\theta \\ (t_{xx}e^{i\varphi_{xx}}-t_{yy}e^{i\varphi_{yy}})\sin2\theta & t_{xx}e^{i\varphi_{xx}}+t_{yy}e^{i\varphi_{yy}}-(t_{xx}e^{i\varphi_{xx}}-t_{yy}e^{i\varphi_{yy}})\cos2\theta \end{array}\right) \end{array}$$
where $R(\theta )=\left(\begin{array}{cc} \cos\theta & \sin\theta \\ -\sin\theta & \cos\theta \end{array}\right)$ is a standard 2 × 2 rotation matrix.

Under normal incidence of CP light, i.e., $E_{in}=\left[\begin{array}{c} 1\\ \pm i \end{array}\right]$, where ‘+’ demonstrates the LCP light and ‘−’ demonstrates the RCP light, the transmitted field can be deduced as

(4)$$\begin{array}{l} E_{out}=T^{*}E_{in}=\frac{1}{2}\left(\begin{array}{c} t_{xx}e^{i\varphi_{xx}}+t_{yy}e^{i\varphi_{yy}}+(t_{xx}e^{i\varphi_{xx}}-t_{yy}e^{i\varphi_{yy}})e^{\pm i2\theta }\\ \pm i(t_{xx}e^{i\varphi_{xx}}+t_{yy}e^{i\varphi_{yy}})\mp i(t_{xx}e^{i\varphi_{xx}}-t_{yy}e^{i\varphi_{yy}})e^{\pm i2\theta } \end{array}\right)\\ \;\;\;\;\;\;\;=\frac{1}{2}\left(t_{xx}e^{i\varphi_{xx}}+t_{yy}e^{i\varphi_{yy}}\right)\left(\begin{array}{l} 1\\ \pm i \end{array}\right)+\frac{1}{2}\left(t_{xx}e^{i\varphi_{xx}}-t_{yy}e^{i\varphi_{yy}}\right)e^{\pm i2\theta }\left(\begin{array}{c} 1\\ \mp i \end{array}\right) \end{array}$$

The transmitted beam includes two parts from Equation (4): one is a co-polarized beam without phase change, and the other is a cross-polarized beam with phase change of ϕ=±2θ, which is known as the PB phase. 

If the nanobrick is a half-wave plate, then $T_{xx}=-T_{yy}$; that is to say, $t_{xx}=t_{yy}$ and $\varphi_{xx}-\varphi_{yy}=\pm \pi$. Then, the transmitted field can be expressed as 

(5)$$E_{out}=T^{*}E_{in}=\frac{1}{2}\left(t_{xx}e^{i\varphi_{xx}}-t_{yy}e^{i\varphi_{yy}}\right)e^{\pm i2\theta }\left(\begin{array}{c} 1\\ \mp i \end{array}\right).$$

More specifically, 

(6)$$E_{RL}=T^{*}E_{L}=\frac{1}{2}\left(t_{xx}e^{i\varphi_{xx}}-t_{yy}e^{i\varphi_{yy}}\right)e^{i2\theta }\left(\begin{array}{c} 1\\ -i \end{array}\right)$$

(7)$$E_{LR}=T^{*}E_{R}=\frac{1}{2}\left(t_{xx}e^{i\varphi_{xx}}-t_{yy}e^{i\varphi_{yy}}\right)e^{-i2\theta }\left(\begin{array}{c} 1\\ i \end{array}\right).$$

That is to say, the transmitted light only contains the cross-polarized light for a half-wave plate. The total phases of the transmitted cross-polarized light can be expressed as $\Phi_{RL}=\varphi_{xx}+2\theta$ and $\Phi_{LR}=\varphi_{xx}-2\theta$, which depend on the structural phase via tailoring of the nanobrick size, the PB phase via rotating of the nanobrick, and the handedness of CP incidences. Furthermore, if the rotated nanobricks simultaneously work for the wavefront corresponding to LCP and RCP incidences, the structural phase of *x*-polarization $\varphi_{xx}$ and rotation angle θ can be expressed as
(8)$$\varphi_{xx}=\frac{1}{2}\left[\left(\Phi_{LR}-2n_{1}\pi \right)+\left(\Phi_{RL}-2n_{2}\pi \right)\right]$$
(9)$$\theta =\frac{1}{4}\left[\left(\Phi_{RL}-2n_{2}\pi \right)-\left(\Phi_{LR}-2n_{1}\pi \right)\right]$$
where $n_{1}$ and $n_{2}$ are integers. Therefore, a dual-phase modulation for LCP and RCP light can be achieved by modulating $\varphi_{xx}$ and θ, which overcomes the spin dependence from the PB phase, improves the efficiency of optical devices, and promotes the integrated development of optical systems.

Here, we design some bi-functional metasurfaces with the RSNs adhered onto SiO_2_ substrate. The complex permittivity of silicon is extracted from data from Pierce [37]. The native oxidation of silicon is self-limiting, and the thickness of the oxide layer is only several angstroms [38]. The effect on optical properties is limited and is generally not considered [39,40,41]. Therefore, the chemical stability of the metasurface based on silicon is very good. As shown in Figure 1a, these RSNs possess a different length of L, width of W, and orientating angles of θ. The period of a unit cell is selected as P = 700 nm. The phase accumulation stems from the waveguide effect related to the height (h) of the RSN, and the height is optimized as 900 nm to achieve the entire 2π phase under incident light with wavelength of 1500 nm. The total phase of the RSNs is composed of the structural phase and PB phase, which can be optimized and verified by the finite-difference time-domain (FDTD) method with the periodic boundary conditions in the x- and y-directions. The structural phase modulation could be realized through tailoring the length (L) and width (W) of the irrotational nanobricks. As shown in Figure 1b,c, for the XLP incidence, the transmittance and the phase of the transmitted XLP light could be expressed as functions of L and W of the rectangular nanobricks, respectively. Figure 1d shows the phase of the transmitted YLP light could also be expressed as the functions of L and W under the YLP incidence. In addition, the modulating structural phase could cover almost the entire 2π with very high transmission efficiency. The PB phase could be achieved by rotating nanobricks. Therefore, the different phase responses for LCP and RCP light can be obtained simultaneously by changing L, W, and θ of the rectangular nanobricks, and they can be manipulated independently. 

## 3. Results and Discussions

To provide proof of our design philosophy regarding bi-functional metasurfaces, we first designed a bi-functional beam deflector to deflect the LCP and RCP light in different directions. It is different from the beam deflector based on the PB phase, which only deflects the LCP and RCP light in opposite directions. On the basis of the generalized form of Snell’s law [42],
(10)$$n_{t}\sin\theta_{t}-n_{i}\sin\theta_{i}=\frac{\lambda_{0}}{2\pi }\frac{d\Phi }{dx}$$
where $n_{i}$ and $n_{t}$ are the refractive index of the incident and refracted space, respectively; $\theta_{i}$ and $\theta_{t}$ are the incident and refractive angles, respectively; and $\lambda_{0}$ is the incident wavelength. When $\lambda_{0}$ and $\theta_{i}$ are constant, the refractive angle depends on the phase gradient of $\frac{d\Phi }{dx}$. Therefore, we selected eight unit cells with different lengths, widths, and rotation angles to satisfy the phase gradient, achieving different deflection angles for LCP and RCP light. This supercell is shown at the bottom of Figure 2a. As shown in Figure 2a, the cross-polarized phase differences were π/2 and π/4 between two neighboring unit cells for LCP and RCP incidences, and the transmittance of the eight unit cells reached nearly 90%. The cross-polarized electric fields are shown in Figure 2b,c for LCP and RCP incidences, respectively, which indicate that the transmitted LCP and RCP light is refracted in different directions. The deflecting angles in the far-field are −32 and −15° for LCP and RCP incidences, respectively, as depicted in Figure 2d, which agrees well with the theoretical angles of −32.4 and −15.5°, implying excellent performance of our designed bi-polarization beam deflector. Because XLP light is composed of LCP and RCP light, this bi-polarization beam deflector can also be regarded as a beam splitter used to steer the LCP and RCP light in different directions. 

Based on the discussion above, we have additionally designed a bi-functional metalens to focus the LCP and RCP light to different positions. In contrast with the multiplexing metalens, our designed bi-functional metalens possess high efficiency and avoid functional crosstalk. The phase of the bi-functional metalens can be expressed as follows:(11)$$\{\begin{array}{c} \Phi_{RL}=\frac{2\pi }{\lambda }\left(\sqrt{{\left(x-x_{0}\right)}^{2}+f^{2}}-f\right)\\ \Phi_{LR}=\frac{2\pi }{\lambda }\left(\sqrt{{\left(x+x_{0}\right)}^{2}+f^{2}}-f\right) \end{array}$$
where λ is the incident wavelength, $x_{0}$ is the off-centered distance of the focal spots, and f is the focal length. We select λ = 1500 nm, $x_{0}$ = 5p = 3500 nm, and f = 7500 nm. According to Equation (11), we select 35 RSNs with different lengths, widths, and rotation angles to achieve symmetrical focusing for the LCP and RCP light, and the numerical aperture can reach (NA) = 0.85, which is relatively high. For the LCP incidence, as shown in Figure 3a, the transmitted RCP light was focused at the preset location of (3.5 µm, 7.5 µm). The efficiency of the metalens nearly reaches 49.8%, and the full width at half maximum (FWHM) is 525 nm for the LCP incidence. For the RCP incidence, as shown in Figure 3b, the transmitted LCP light is also focused at the ideal location of (−3.5 µm, 7.5 µm). The efficiency of the metalens is about 50.1%, and the FWHM is 520 nm. That is to say, the designed bi-functional metalens possesses spin-selected function and high focusing quality. XLP light has been known as the combination of the RCP and LCP light, and as shown in Figure 2c, there also are two focuses in the preset positions for LCP and RCP lights. Therefore, our designed bi-functional metalens is also a bi-focusing metalens for the LP incidence. We should note here that the concrete parameters, such as the focusing length, the focusing positions, and so on, can be manipulated easily in the designing process. 

We also designed a bi-functional metasurface with the combined spin-dependent focusing and beam deflection functions (BMFD), as shown in Figure 4a. This BMFD can be considered a metalens for the LCP incidence, with the phase distributions along the x-axis expressed as follows:(12)$$\Phi_{RL}=\frac{2\pi }{\lambda }\left(\sqrt{{\left(x-x_{0}\right)}^{2}+f^{2}}-f\right)$$
where the incident wavelength λ = 1500 nm, and the focal length f = 7.5 μm. In the numerical simulation, periodic boundary conditions were employed in the y-direction, and perfectly matched layer boundary conditions were used in the z-direction. The supercell is made up of 32 RSNs, and possesses a parabolic phase profile for the LCP incident light, as shown in Figure 4a, in which NA = 0.83. Meanwhile, this metasurface can also be regarded as a beam deflector for the RCP incidence, in which eight RSNs can achieve a linearly varying 2π phase shift, and can be seen as a sub-supercell, as shown in Figure 4a, and four sub-supercells make up the supercell. The concrete calculated phase distributions for the LCP and RCP incidences are shown in Figure 4b. For LCP incidence, Figure 4c shows the perfect focusing performance of transmitted RCP light, which is focused at 7.5 μm behind the metalens. The FWHM is 534 nm, as shown in Figure 4c, and the focusing efficiency of the BMFD is about 49.9%, which implies the designed BMFD has relatively superior focusing performance under LCP incident light. For RCP incidence, Figure 4d shows the electric field distribution of the transmitted LCP light, and the deflecting angle is approximately −15°, which agrees well with the theoretical angle of −15.5°, as shown in Figure 4e, which demonstrates that the designed BMFD possesses excellent deflection characteristics under RCP incident light. Therefore, our designed BMFD can achieve spin-dependent focusing and beam deflection effectively. It should be noted that the concrete parameters of the designed BMFD, such as the focusing length, the deflection angles, and so on, can be manipulated easily in the designing process.

To demonstrate the universality and validity of our designing method, we also designed a bi-functional vortex phase plate (BVPP) which can produce a vortex beam with different topological charges for LCP and RCP incidences. To design the BVPP, we selected eight nanobricks as used in the bi-functional beam deflector, as shown in Figure 5a, and the phase accumulations in a circle are 4π (the concrete modulating phases are −π, −π/2, 0, π/2, π, 3π/2, 2π, and 5π/2), and 2π (the concrete modulating phases are −π, −3π/4, −π/2, −π/4, 0, π/4, π/2, and 3π/4) for LCP and RCP incidences, respectively. The designed metasurface consists of 14 × 14 nanobricks, and the interface is divided into eight parts arrayed around the center of the BVPP with a phase increment. For LCP incidence, a phase shift of 4π was achieved around the center of the metasurface, which produced a vortex beam with a topological charge of l = 2. The phase profile 7000 nm away from the BVPP is presented in Figure 5b, which indicates the vortex beam produced with l = 2. For RCP incidence, a phase shift of 2π was achieved around the center of the metasurface, which produced a vortex beam with a topological number of l = 1. The phase profile 7000 nm away from the BVPP is presented in Figure 5c, which indicates that the vortex beam produced with l = 1. It should be noted that the concrete topological charges (including the signs and the numbers) of the designed BVPP can be manipulated easily by changing the phase distributions accordingly.

The designed bi-functional metasurfaces possess multispectral characteristics with a wavelength range of 1200−1700 nm, and the fractional bandwidth reaches up to 34%. To verify this broadband feature of our designed metasurface, we focused our research on the BMFD with focusing and deflection functions. For LCP incidence, the focal lengths reduced quickly with an increase in wavelength, as shown in Figure 6a. To quantitatively study the focusing quality, the electric field intensity at the focal plane along the x-direction under different incident wavelengths is presented in Figure 6c. The largest intensity and smallest FWHM were observed for the wavelength of 1500 nm, and there are still good focusing qualities for other wavelengths, despite the lower intensities and larger FWHMs. For RCP incidence, Figure 6b shows the electric field distribution under different incident wavelengths, which implies that the deflecting angles vary with incident wavelength. Figure 6d shows that the deflecting angle increases slowly with an increase in wavelength, and the deflecting angle approximately matches the theoretical value from the generalized form of Snell’s law. Therefore, we can also say that the designed bi-functional metasurface exhibits outstanding broadband characteristics in the near-infrared band, and that it will possess the largest efficiency for the designed wavelength of 1500 nm. The broadband characteristics of the designed devices (beam splitter, metalens, BVPP, and single unit cell) can be found in Appendix A.

## 4. Conclusions

In conclusion, we have demonstrated four spin-dependent bi-functional metasurfaces in the near-infrared band, which are composed of a series of RSNs of varying sizes and orientations. These RSNs are skillfully designed based on the structural phase and PB phase simultaneously. The specific designing method and the transmitted matrix of each metasurface are described and deduced strictly, which opens a new degree of freedom for the bi-functional metasurfaces. We have designed a bi-functional beam deflector, which can deflect LCP and RCP light in different directions. We then designed a bi-functional metalens, which can focus the LCP and RCP light to different positions. We have also designed a BMFD which can achieve the focusing and deflection functions for LCP and RCP incidences, respectively. We finally designed a bi-functional BVPP, which can produce a vortex beam with different topological charges for LCP and RCP incidences. This research on broadband spin-dependent bi-functional metasurfaces paves the way for future applications in integrated optics and spin-based optical devices.

## Figures and Tables

**Figure 1 nanomaterials-09-01744-f001:**
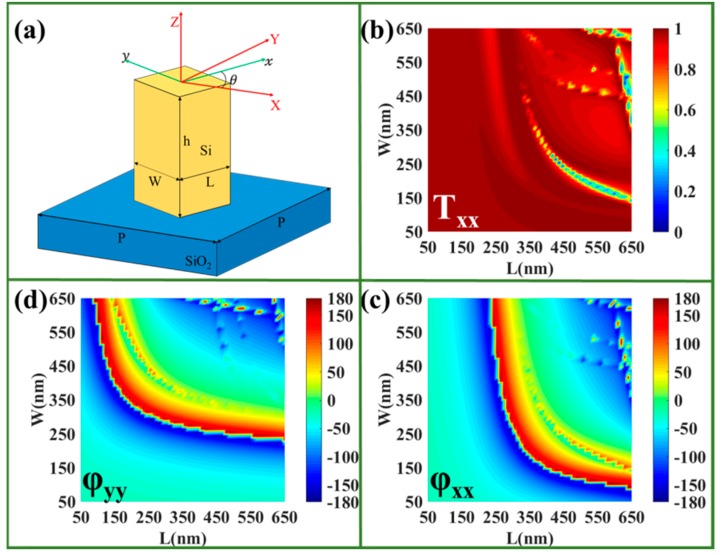
(**a**) Schematic of rotary silicon nanobrick (RSN) with length *L*, width *W* and orientation θ. The x-linearly polarized (XLP) (**b**) transmittance and (**c**) phase of the irrotational Si nanobricks as a function of *L* and *W* with XLP incidence. (**d**) The phase of the transmitted y-linearly polarized (YLP) light as a function of *L* and *W* with YLP incidence (the color scale units of transmittance intensity (**b**) and phase (c,d) are ${\left(V/m\right)}^{2}$ and °, respectively).

**Figure 2 nanomaterials-09-01744-f002:**
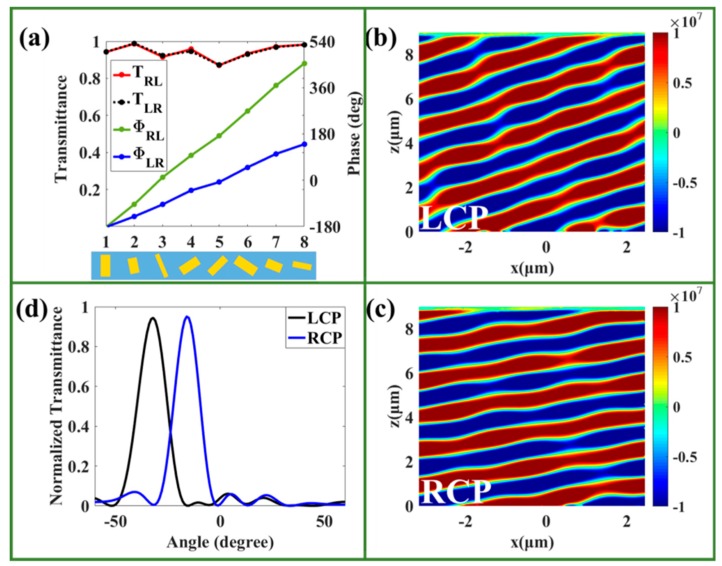
(**a**) The transmittance and phase of transmitted light through the selected eight RSNs for left circularly polarized (LCP) and right circularly polarized (RCP) incidences with wavelength of 1500 nm. The electric field distributions of the bi-functional beam deflector under (**b**) LCP and (**c**) RCP incidences (the color scale unit of the electric field (**b**,**c**) is V/m). (**d**) The normalized transmittance serves as a function of scattered angles under LCP and RCP incidences.

**Figure 3 nanomaterials-09-01744-f003:**
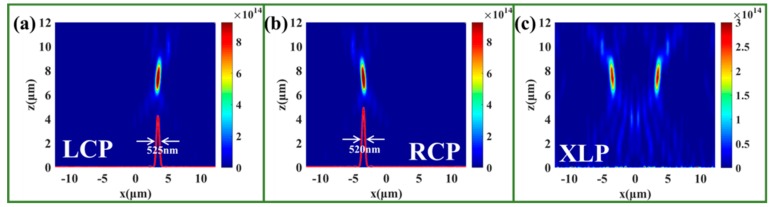
The intensity distributions of the bi-functional metalens under (**a**) LCP, (**b**) RCP, and (**c**) XLP incidences. The red curve is the intensity along the x-axis at the focusing plane (the color scale unit of the intensity (**a**–**c**) is ${\left(V/m\right)}^{2}$.

**Figure 4 nanomaterials-09-01744-f004:**
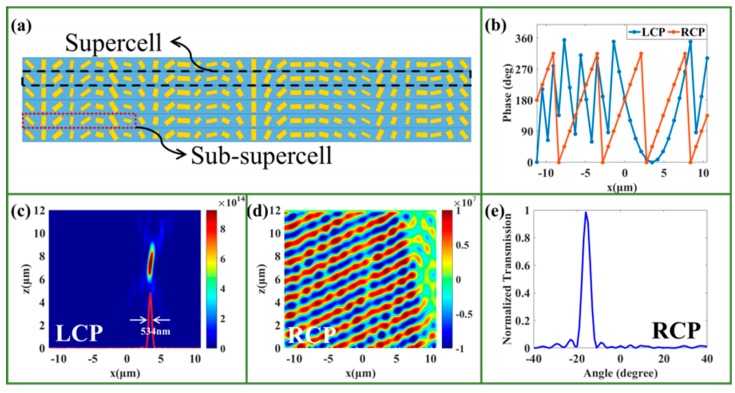
(**a**) Schematic of designed beam deflection functions (BMFD). (**b**) Phase distributions of the BMFD under LCP and RCP incident light, respectively. (**c**) The intensity distributions of the BMFD under LCP incident light. (**d**) The electric field distributions of the BMFD under RCP incident light (the color scale units of the (**c**) intensity and (**d**) electric field are ${\left(V/m\right)}^{2}$ and V/m, respectively). (**e**) The transmittance serves as a function of scattered angles under RCP incident light.

**Figure 5 nanomaterials-09-01744-f005:**
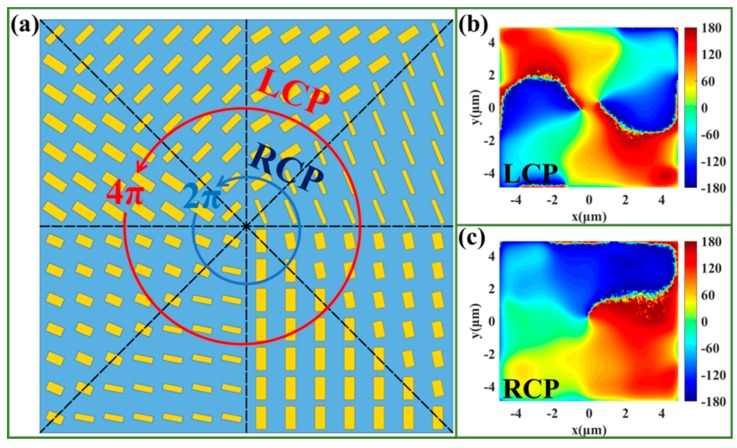
(**a**) Sketch map of the designed bi-functional vortex phase plate (BVPP). The phase distributions of the produced vortex beam under (**b**) LCP and (**c**) RCP incidences (the color scale unit of the phase (**b**,**c**) is °).

**Figure 6 nanomaterials-09-01744-f006:**
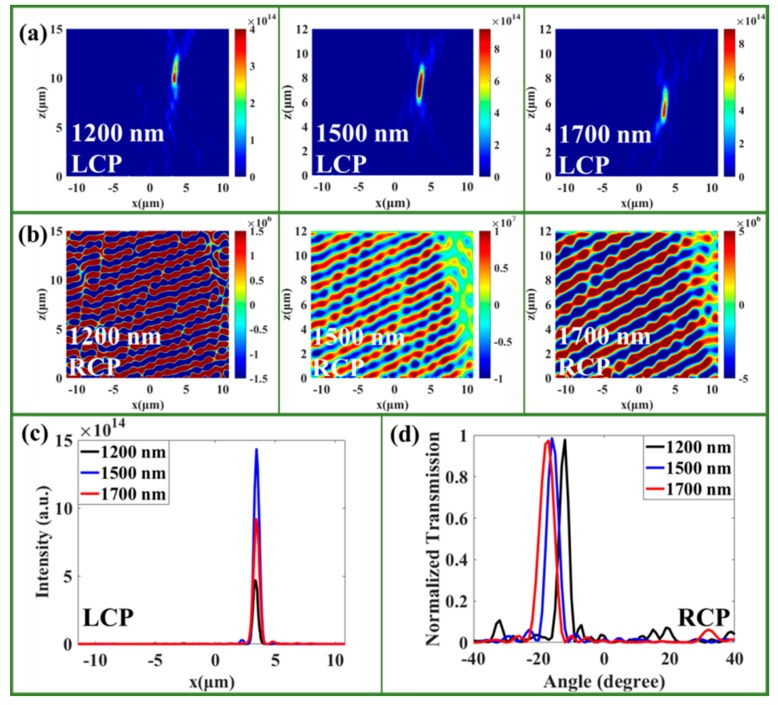
(**a**) Intensity distribution of the BMFD under LCP incident light at the incident wavelengths of 1200, 1500, and 1700 nm. (**b**) Electric field distribution of the BMFD under RCP incidence at the incident wavelengths of 1200, 1500, and 1700 nm (the color scale units of the (**a**) intensity and (**b**) electric field are ${\left(V/m\right)}^{2}$ and V/m, respectively). (**c**) The electric field intensity along the x-axis at the focusing plane under LCP incident light with different wavelengths. (**d**) The transmittance serves as a function of scattered angles under RCP incident light with different wavelengths.

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
