# Peer review of "High-Efficiency and Broadband Near-Infrared Bi-Functional Metasurface Based on Rotary Different-Size Silicon Nanobricks"

_nanomaterials, 2019, doi:10.3390/nano9121744_

Round 1

Reviewer 1 Report

In this manuscript, the authors demonstrate four spin-dependent bi-functional dielectric metasurfaces in the near-infrared band, through combining the structural phase and PB phase simultaneously. Four metasurfaces possess different functions and designing philosophies. And the transmission mechanism is also deduced in details, so that the readers can understand and grasp the designing methods very well. In additions, the wide-band characteristic of the metasurface has also been investigated and proved in the paper. The results demonstrate the superior performances of the designing devices. The work is of significance for the development of high-efficiency bi-functional devices without crosstalk.

This work is worth to be published and I recommend publication of this manuscript in Nanomaterials.

There are some minor comments/questions, which could improve the present manuscript.

The parameter of silicon is not given in the whole manuscript, while I think it should have influence on the transmission coefficients. Therefore, the authors should give the related parameters clearly. The authors get Equations 8 and 9 by analyzing a unit cell, which act as a half-wave plate (Txx = -Tyy). It would be helpful for understanding by giving out a map about in the Figure 1. Please tell us more about how to select the corresponding structure. Besides, the BMFD with the combined spin-dependent focusing and beam deflection functions is intriguing, the concrete focusing efficiency should be given to verify its performance.

Reviewer 2 Report

see attached file

Reviewer 3 Report

The manuscript “High-Efficiency and Broadband Near-Infrared Bi-Functional Metasurface Based on Rotary Different-Size Silicon Nanobricks” is well presented and scientifically interesting. Only few minor changes regarding Figures are necessary.

-Concerning all the Figures of the manuscript, please adjust the positions of the letters of the Figures (e.g., (a), (b) and (c)); specifically, the position of the letters should more distant from the numbers of the graph axes.

-Figure 6c, please add “(a.u.)” on the y axis, if necessary.

-Figures 6a, 6b, 6c, 6d, please add a space between values and units (nm).

-Figures 1b, 1c, 2b, 2c, 3a, 3b, 3c, 4c, 5b, 5c, 6a and 6b, please specify, when necessary, the units or the arbitrary units associated to the color scale (only numbers are shown associated to the color scales).

Reviewer 4 Report

The manuscript titled “High-Efficiency and Broadband Near-Infrared Bi-Functional Metasurface Based on Rotary Different-Size Silicon Nanobricks” presents a metasurface that combines Pancharatnam-Berry (PB) and structural phase to achieve bi-functional designs, such as lenses, beam-steerers or vortex generators, with different response depending on the handedness of a circular polarized wave at the input. In this regard, this can be seen as a new example of similar structures exploiting the same principle, such as the ones reference by the authors (Refs. 33 and 34). Therefore, I find that the content might be considered incremental, as the structure used is quite usual in the context of PB metasurfaces (a dielectric fin) and in particular in metasurfaces combining PB and structural phase. This is my main criticism to the work. I have also other comments that could be addressed:

As stated by the authors, the main novelty of the work seems to be the relatively broad band operation. However, this should be quantified more firmly, beyond some simulations results corresponding to the lens. Simulation results showing the broadband response of the unit cell elements (both magnitude and phase) should be included in the manuscript. In particular, each design (lens, beam steerer, vortex generator) should have the response of the different unit cells employed. The operation bandwidth should be put in terms of the functionality pursued. In addition, this should be put in numbers using for example the fractional bandwidth. In this regard, it is worth commenting that the maximum fractional bandwidth in the present design is around 34% (bandwidth of 500 nm and central wavelength of 1450 nm). In a previous work [Benedikt Groever, Noah A. Rubin, J. P. Balthasar Mueller, Robert C. Devlin & Federico Capasso, “High-efficiency chiral meta-lens” Scientific Reports volume 8, Article number: 7240 (2018)] a lens designed following the same principles achieved a similar figure of 32% (bandwidth of 180 nm for a central wavelength of 560 nm). In that work, a experimental proof was included.

All in all, I find the work incremental. The claimed broad band operation should be justified, quantified and compared with the literature. As it is now, I cannot accept its publication.

Round 2

Reviewer 4 Report

I have read the revised version as well as the Authors’ response carefully. After this assessment, I still find that the proposed research can be considered incremental. The paper lacks experimental results and there are now papers in the literature that do have measurements. In addition, the broadband operation is not firmly demonstrated. I do not agree with the authors response, stating that a single device (the lens) is representative of all. In fact, the unit cell combinations are different. Therefore, the coupling between unit cells is different for each prototype and could reduce the operation bandwidth in some cases. To be sure about the actual operation bandwidth, each device should be analyzed separately. Finally, I still think that it is important to see the simulation results in magnitude and phase of each unit cell. I am aware that there are numerous cells but it is quite usual in the specialized literature dealing with PB metasurfaces to put this information. It could be included as Supplementary Information.

Author Response

See the Attached files!
